# Task Demands and Sentence Reading Comprehension among Healthy Older Adults: The Complementary Roles of Cognitive Reserve and Working Memory

**DOI:** 10.3390/brainsci13030428

**Published:** 2023-03-01

**Authors:** María Teresa Martín-Aragoneses, Gema Mejuto, David del Río, Sara Margarida Fernandes, Pedro F. S. Rodrigues, Ramón López-Higes

**Affiliations:** 1Department of Research Methods and Diagnosis in Education II, National University of Distance Education (UNED), 28040 Madrid, Spain; 2Vianorte-Laguna Foundation (FVN-Laguna), 28047 Madrid, Spain; 3Department of Experimental Psychology, Complutense University of Madrid (UCM), 28223 Madrid, Spain; 4Centre for Cognitive and Computational Neuroscience, Complutense University of Madrid (UCM), 28223 Madrid, Spain; 5Portucalense Institute for Human Development (INPP), Portucalense University (UPT), 4200-072 Porto, Portugal

**Keywords:** cognitive reserve, older adults, sentence comprehension, sentence–picture verification, task demands, working memory

## Abstract

Ageing entails different functional brain changes. Education, reading experience, and leisure activities, among others, might contribute to the maintenance of cognitive performance among older adults and are conceptualised as proxies for cognitive reserve. However, ageing also conveys a depletion of working memory capacity, which adversely impacts language comprehension. This study investigated how cognitive reserve proxies and working memory jointly predict the performance of healthy older adults in a sentence reading comprehension task, and how their predictive value changes depending on sentence structure and task demands. Cognitively healthy older adults (*n* = 120) completed a sentence–picture verification task under two conditions: concurrent viewing of the sentence and picture or their sequential presentation, thereby imposing greater demands on working memory. They also completed a questionnaire on cognitive reserve proxies as well as a verbal working memory test. The sentence structure was manipulated by altering the canonical word order and modifying the amount of propositional information. While the cognitive reserve was the main predictor in the concurrent condition, the predictive role of working memory increased under the sequential presentation, particularly for complex sentences. These findings highlight the complementary roles played by cognitive reserve and working memory in the reading comprehension of older adults.

## 1. Introduction

Language processing and working memory (WM) are closely linked in our everyday lives. For example, consider a situation where you are trying to cook a dish by following the steps of a recipe video on YouTube. You have been cautious about preparing all the ingredients in advance. In such a situation, WM will work as a central workspace where the meaning extracted from the instructions and the visual information gathered from the environment are held and compared in order to guide your actions. Now suppose that you need a special kind of bowl but are unable to find it in the kitchen, so you pause the video, get out, and go ask someone in your family about where the bowl might be. After receiving some instructions on this, you come back to the kitchen and use those instructions to search for the bowl. In this situation, you are also using WM to integrate the previous verbal information and the visual information from the environment, but now you are relying more heavily on the adequate encoding, maintenance, and retrieval of verbal information in order to successfully perform the task. While a slight reduction in WM resources might not have a significant effect in the first scenario, it might have a very negative effect in the second one.

The interaction between WM and language processing might be particularly relevant for older adults given that while basic linguistic processing does not seem to be affected among cognitively healthy older adults, the decrease in WM capacity with advancing age might adversely affect the performance of complex tasks involving language comprehension [1,2,3]. In this regard, there is evidence for the upregulation of WM and executive brain networks among older adults during complex language comprehension, which is possibly associated with compensatory processing owing to the deleterious effects of old age on brain networks [4,5].

Moreover, there appear to be certain neuroprotective factors that might contribute to maintaining high cognitive performance as a person ages. Education, reading habits, and occupation and leisure activities have been identified as factors contributing to the better maintenance of cognitive status among older adults [6,7,8,9,10]. All these individual characteristics have been proposed as proxies for the construct of the cognitive reserve (CR). The basic idea behind the concept of CR is that individual differences in the organisation of cognitive brain networks might allow some people to cope better than others with regard to detrimental changes in brain functioning, including those related to healthy ageing. Cognitive reserve could also be regarded not only as a protective construct in the case of age-related decline or pathology. It can also be viewed as an effect of cumulative experience to foster cognitive performance in healthy older adults [11,12,13].

In recent years, numerous studies have shown a relationship between CR and cognitive function among older adults (e.g., [14,15]; for meta-analytical reviews, see also [16,17]). For instance, CR, operationalised as variance in neuropsychological memory test scores after accounting for sociodemographic factors and brain pathology, has been associated with a lower decrease in reading and language skills over a three-year period, along with a lower likelihood of meeting the criteria for mild cognitive impairment or conversion to dementia [18]. Likewise, CR, measured as a proxy composite score that considers education, occupation, reading habits, and other sociodemographic and lifestyle-related factors, has been associated with better language comprehension through the mediating role of executive function among healthy older adults [19]. Montemurro et al. [20] found that higher CR contributed not only to faster word-by-word reading times in older adults but also to a faster resolution of syntactic violations. This suggests a more efficient online linguistic processing with higher CR, although a similar effect was not detected in the case of semantic violations. Therefore, increased CR might boost language processing. Moreover, it might play a protective role by compensating for age-related decline in cognitive skills such as WM. For example, Payne et al. [7] found that older individuals with higher levels of reading experience demonstrated a reduced effect of WM on online measures of reading difficulty. However, this beneficial role of CR might also depend on how heavily comprehension tasks rely on WM demands. While the performance might depend more on CR proxies such as education or reading habits and less on WM capacity when WM demands are low, the compensatory role played by CR might become weaker when older people are forced to maintain verbal information for a short amount of time before they are allowed to respond.

Furthermore, syntactic complexity (i.e., the adjustment of syntactic constituents to a canonical word order) or propositional density (i.e., the quantity of semantic information that a sentence conveys) would increase language-processing demands at the sentence level. Sentences with a canonical word order, as in English or Spanish subject-verb-object (SVO) sentences, are easier to process and understand than object-before-subject sentences [21,22]. Similarly, multiclausal sentences, which include two or more semantic propositions, as in “The dog that bit the horse is big”, also increase language-processing demands. Importantly, the aforementioned sentence simultaneously introduces information on two events: (a) the dog bit the horse, and (b) the dog is big. Among older adults, sentence comprehension is adversely affected by these factors, particularly in relation to WM depletion [23,24]. López-Higes et al. [25] found that older adults with higher scores in CR proxies were less affected by noncanonical word order and propositional density.

Waters et al. [26] compared the performance of a group of people with mild or moderate Alzheimer-type dementia with that of a group of cognitively healthy older adults in a sentence-to-picture matching task, with pictures presented either before or after the sentence was orally presented. They found no consistent differences between when the pictures were presented before the sentences and when they were presented after sentence processing. While this might call into question whether holding verbal information in memory significantly increases the WM demands among older adults, their study did not consider how interindividual variability related to WM capacity or CR might assist older adults in coping with task demands.

To the best of our knowledge, no study has addressed how WM capacity and CR might separately and jointly influence the comprehension of sentences with different levels of complexity and how this might change depending on whether linguistic information can be immediately used for task accomplishment or whether it has to be held in memory for a short period of time.

Therefore, the three aims of the present study were as follows:

1. We aimed to examine the effect of a short delay on the performance of older adults in a sentence comprehension task. In order to do so, we used a written sentence–picture verification task, which involved making a comparative decision on the basis of whether the interpretation of a written sentence matched a visually presented picture [27]. We compared a concurrent version of the task, where both stimuli (i.e., sentence and picture) were simultaneously present, with a serial version of the task, where the picture appeared only after the participant had read the sentence. Therefore, in the concurrent condition, both stimuli were presented simultaneously when being compared, and the participants could reread the sentence if necessary. On the contrary, in the serial condition, this comparison required the temporary maintenance, in an active and accessible state, of the memory representation generated for the sentence because the participants were not allowed to reread it.

2. We aimed to assess the explanatory value of CR proxies and WM capacity in the performance of older adults in this task, according to the presentation modality (i.e., concurrent vs. serial).

3. Finally, we aimed to explore whether the role of CR proxies and WM capacity in the concurrent and serial modalities also changed depending on the linguistic complexity of the stimuli, by manipulating the adjustment to canonical SVO word order and the propositional density of the sentences.

Given the available evidence, we expected CR proxies and WM to play positive roles in the performance of older adults. According to the findings of the study conducted by Waters et al. [26], the comparative results pertaining to the effect of a serial versus concurrent presentation were uncertain; however, we expected the performance to be modulated by WM capacity more in the serial condition than in the concurrent condition of the task. Furthermore, we also hypothesised that CR proxies might to a greater extent modulate the performance on the sentence–picture verification task when WM demands were low, i.e., in the concurrent version and sentences with lower complexity, while it was expected to be more dependent on WM capacity for the serial version of the task, especially as linguistic complexity increased.

## 2. Materials and Methods

### 2.1. Participants

The incidental sample consisted of 120 Spanish-speaking older adults between the ages of 60 and 80 with no signs of cognitive impairment or subjective memory complaints (suggested as the earliest sign of Alzheimer disease), who were recruited through advertisements in several sociocultural centres for older people in Spain.

The participants were excluded from the experimental part of the study if they (a) scored below 27 on the mini-mental state examination (MMSE [28]; Spanish adaptation by Lobo et al. [29]), (b) scored higher than 5 on the abbreviated version of the geriatric depression scale (GDS-15 [30]), (c) scored higher than 13 on the memory failures of everyday questionnaire (MFE [31]; Spanish adaptation by Montejo-Carrasco et al. [32]), (d) had uncorrected sensory deficits, (e) reported a history of psychiatric or neurological illness, and/or (f) presented diseases that have a high risk of causing cognitive deficits.

Moreover, the participants were randomly assigned to one of two groups, depending on the version of the sentence–picture verification task that they were to perform (i.e., concurrent vs serial). Both groups consisted of 60 older adults. The group assigned to the concurrent condition was composed of 26 men and 34 women, while 24 men and 36 women constituted the serial condition group. Importantly, the groups did not differ in terms of the male/female ratio [χ^2^_(1)_ = 0.137, *p* = 0.711]. As Table 1 shows, the groups also did not differ in relation to age, overall cognitive status, self-reported depressive symptomatology, and subjective complaints of cognitive decline. Table 1 also reports descriptive data and group differences in the sentence comprehension test (ECCO_Senior), along with each one of its subparts (see Section 2.2.3).

### 2.2. Materials

Apart from the aforementioned screening tests, the following additional measures were employed.

#### 2.2.1. The Cognitive Reserve Questionnaire (CRQ)

The CRQ [9] provides an estimate of CR from a composite measure of different proxies. This questionnaire covers 8 areas that have been associated with the formation of CR: education, parents’ education, continuing education, occupation, musical training, knowledge of co-official and/or foreign language(s), reading activity, and engagement in intellectual games. The maximum score is 25. Studies that have addressed the psychometric properties of the CRQ have found evidence of unidimensionality, both in the general population of older adults [33] and in people with Alzheimer disease [34], even if musical training and engagement in intellectual games tended to load lower on the general factor [33]. These studies have also shown adequate reliability, with Cronbach’s alpha values around 0.8 and a categorical omega reliability coefficient of 0.72. Different studies have shown an association of CR with measures of cognitive performance across several domains, such as episodic memory, working memory, attention/executive functions, and overall cognitive performance [9,15,19,33].

In this study, the participants’ scores on the CRQ varied from 3 to 24, with a mean score around 14, for a maximum of 25 points. For reference, the original report [9] places scores of 6 or below in the lowest quartile of their sample; scores between 7 and 9 and between 10 and 14 in the second and third quartiles, respectively; and scores of 15 and above in the higher quartile.

#### 2.2.2. Digit Ordering (DO)

This task, developed by MacDonald et al. [35] to measure WM capacity, consists of listening to a series of unordered digits (e.g., 7-2-4-6-5), which are presented at a rate of one digit per second, for its immediate recall in ascending order (i.e., 2-4-5-6-7). The version used for the study was composed of 15 series, the length of which increased by one digit every three trials, beginning with a two-digit span and increasing up to a maximum length of six. The task ended when the participant was not able to recall at least two sequences of a particular span. The scoring was based on the number of series correctly recalled (from 0 to 15), given that there is evidence that this score may be more sensitive to individual differences [36].

As shown in Table 1, the performance in the DO task oscillated between a score of 7/15 correct trials (which means that only around a 47% of trials was correctly recalled) and 15/15, with a mean score around 12/15 (80% of trials correctly recalled). This reflects a wide variability in performance across participants from both groups. Furthermore, the performance observed in the present study is comparable to that reported by MacDonald et al. [35] for their group of cognitively healthy older adults (mean correct percentage = 83.1%). In addition, note that young adults from MacDonald et al.’s study showed a mean percentage of correctly recalled trials of 90.5% [35].

#### 2.2.3. The ECCO_Senior Test

The ECCO_Senior test (exploración cognitiva de la comprensión de oraciones para mayores [37,38]) is a short test consisting of 36 sentence–picture pairs conceived to assess the reading comprehension of semantically reversible sentences among older adults through a verification task. These sentences use a high-frequency vocabulary. Depending on the possible relationships that can occur between the sentence and the picture, there are three types of items: congruent items (where the picture exactly represents the meaning of the sentence), lexical foils (where either a visual element or the action depicted does not match what the sentence describes), and syntactic foils (where thematic roles are reversed with respect to what the sentence expresses). Figure 1 presents an example for each of these types of items. On the other hand, the test is composed of 12 Spanish sentence structures (see Table 2), with three items (a congruent item, a lexical foil, and a syntactic foil) for each one of them. Therefore, one-third of the test items implied a positive response (i.e., true). Because Spanish allows for a greater freedom in word order than English or other languages, this test explores a variety of syntactic constructions. Moreover, these syntactic constructions could be grouped according to two orthogonal factors related to sentence complexity. Specifically, half of the sentences display a canonical Spanish SVO word order (canonical sentences: CSs), while the remaining demonstrate a noncanonical word order (noncanonical sentences: NoCSs). Additionally, half of them contain just one proposition (1PS), while the remaining are formed by two propositions (2PSs). Table 2 summarises the different sentence structures classified according to their adjustment to canonical word order and their propositional density. The score was equal to the number of correct responses, with a maximum of 36 points for the overall test and 18 for each one of the four conditions (CS, NoCS, 1PS, and 2PS).

Two versions of the task were employed in this study: a concurrent version, in which sentence and picture were presented together, and a serial version, in which the picture that accompanied each sentence was shown immediately after the participant had finished reading the sentence. In both versions, the items were displayed on a computer screen, and the participants were asked to read the sentences aloud at their own pace and then give their responses regarding the correspondence between the sentence and the picture by indicating whether it was true/false with no time constraints. In the concurrent version, both stimuli (i.e., sentence and picture) simultaneously appeared and remained on the screen until a response was given while allowing the participants to reread the sentence if necessary. In the serial version, the participants first read the sentence shown on the screen and then pressed a button on the keyboard to view the picture, which was present until a response was given. In this latter version, the presentation of the picture caused a permanent disappearance of the sentence, without being able to go back to it for rereading. Each participant performed the task in only one of these two conditions.

### 2.3. Procedure

First, the older adults who expressed an interest in participating in the study received an information sheet and a consent form. After providing their informed consent to participate in the study, they were requested to provide basic demographic data as well as information on their medical history and health status. Next, screening tests were conducted. For each participant, we checked to ensure that none of the exclusion criteria had been met. They were then randomly assigned to one of two groups formed, on the basis of the version (i.e., concurrent vs serial) of the sentence–picture verification test that they took. The participants subsequently completed the ECCO_Senior test according to the group assigned. The session ended with the application of the CRQ and the DO task. All the instruments were administered in a single individual session by qualified and trained personnel. This session lasted approximately 45–60 min, with the possibility of taking a break, although this was not found to be necessary. A consultation with the primary care physician was recommended whenever an alarm sign was suggested by the scores on the screening tests. This research was carried out with the approval of the corresponding ethics commission.

### 2.4. Analysis

The univariate normality was checked through the skewness and kurtosis values for each continuous variable of the study. As shown in Table 1, the values of these indices were within the acceptable range of ± 2, which were even lower than |± 1.5| for most of the variables [39]. The multivariate normality was verified by using Mardia’s test (*p*-values > 0.298) [40]. The confirmation of these assumptions made choosing parametric tests possible.

The differences between the groups were analysed by using the chi-square test for the categorical variable *sex* and Student’s *t*-tests for independent samples in the case of continuous variables. Moreover, Student’s *t*-tests for paired samples were used to determine intragroup differences. The effect size of the significant differences between the conditions was estimated by using Cohen’s *d* statistic [41], and the estimates of the intraindividual effect size were calculated by using the pooled standard deviation, controlling for the intercorrelation of both measures. These estimates are reported here in absolute values.

Furthermore, Pearson’s product-moment correlation coefficient was used to determine the degree to which the performance of the participants in the sentence–picture verification test was related to the CR and WM scores and to the rest of the variables. In addition, separate stepwise multiple linear regression analyses were conducted to determine significant predictors and quantify their explanatory value. To confirm the validity of the models, the independence of the residuals was tested by using the Durbin–Watson *D* statistic. The *D* values of around 2 were seen as indicative of the absence of autocorrelation [42]. Additionally, the compliance with the assumption of noncollinearity between the independent variables of the regression models was verified by using the tolerance level (TOL) and the variance inflation factor (VIF), values close to 1 indicating the independence of the variables. Both bivariate correlation analyses and multivariate analyses were independently run for each group.

In order to gain a better understanding of the predictor variables, communality analyses were carried out through a series of simple and multiple linear regression analyses where the overall *R*^2^ of the criterion variable was partitioned into portions of the unique and shared variance of the set of predictor variables. The results are presented through Venn diagrams in order to facilitate their visualisation.

All the statistical analyses were carried out with the help of the SPSS version 24 statistical programme, while the different effect sizes were estimated by using Psychometrica online calculators [43].

## 3. Results

### 3.1. Concurrent vs. Serial Presentation

According to the concurrent and serial groups’ scores on the CRQ and the DO task, the groups were equivalent in terms of CR and WM (see Table 1). In terms of sentence reading comprehension, all the participants performed above chance on the ECCO_Senior test, regardless of the version of the sentence–picture verification task completed. Likewise, significant differences between the men and the women were ruled out both in the concurrent condition (*t*_(58)_ = 1.074, *p* = 0.287) and in the serial condition (*t*_(58)_ = −0.094, *p* = 0.925), making it unnecessary to include sex as a covariate in the analyses. However, the group assigned to the concurrent condition obtained a significantly higher overall average score on the ECCO_Senior sentence–picture verification task than the group assigned to the serial condition (*t*_(118)_ = 2.179, *p* = 0.031, *d* = 0.40).

### 3.2. Role of CR and WM in the Concurrent and Serial Presentation

As displayed in Table 3, a strong and positive association was observed between the performance of both groups on the ECCO_Senior test and the CRQ and DO scores, which also significantly correlated with each other. Age was not significantly related to performance in either of the two versions of the ECCO_Senior test. In contrast, age did significantly and negatively correlate in both groups with the general cognitive functioning of the participants, as expressed by the score on the MMSE. On the other hand, the MMSE score positively and significantly correlated with the scores on the CRQ and the DO task in both groups. Similarly, a positive and significant relationship was observed between the MMSE score and the performance on both conditions of the ECCO_Senior test. The MFE score was not correlated with participants’ performance in the ECCO_Senior sentence–picture verification task.

Only in the group assigned to the serial condition did age significantly correlate with the score on the DO task, although in both groups, the association between these two variables was negative. Moreover, an inverse relationship was found in both groups between the score obtained in the GDS-15 and all the other variables of the study, apart from the MFE scores, although the GDS-15 scores were significantly associated only with the CRQ and ECCO_Senior scores in the group assigned to the concurrent condition.

Given the results obtained in the correlation study, all variables were included in the regression analyses. Age did not explain any significant portion of the variance in the regression analyses in either the concurrent version or the serial version of the ECCO_Senior test for any of the regression models. The same was true of the other global measures: MMSE, GDS-15, and MFE.

A single significant model emerged from the regression analysis for the concurrent condition, where CR accounted for around 30% of the total variability of the criterion variable (see Table 4). In contrast, two significant models were obtained for the serial condition (see Table 4). The first model (i.e., Model 1) also identified CR as the main explanatory variable, which accounted for around 23% of the variance in the dependent variable, while Model 2 included WM, which added 9% of additional explained variance. Hence, CR was the only variable with explanatory power in both conditions (i.e., concurrent and serial). The main difference between them was in the specific contribution made by WM, which explained the additional variance in the serial condition.

Figure 2 represents in detail the unique and shared contributions made by CR, WM, and all the rest of the variables (i.e., age, cognitive status, specific depressive symptoms, and self-perception of cognitive functioning) in each ECCO_Senior condition. Importantly, the proportion of variance explained by these Venn diagrams is slightly higher than that reported by the models obtained from the stepwise regression analyses. This is because these representations also include the proportions of variance explained by the variables that do not significantly contribute in a specific way. Also noteworthy is that the specific variance explained by CR and WM capacity is higher in the serial version, especially for WM capacity.

### 3.3. Linguistic Complexity for Concurrent and Serial Versions and the Role of CR and WM

As expected, the noncanonical sentences were harder to process than the canonical sentences (see Table 1) in both the concurrent (*t*_(59)_ = 9.330, *p* < 0.001, *d* = 1.23) and the serial (*t*_(59)_ = 7.378, *p* < 0.001, *d* = 0.88) versions of the ECCO_Senior test. Similarly, the sentences containing two propositions were harder than the sentences consisting of one (see Table 1) for the concurrent (*t*_(59)_ = 3.673, *p* = 0.001, *d* = 0.43) and serial (*t*_(59)_ = 4.359, *p* < 0.001, *d* = 0.56) versions.

When considering the effect of serial compared to concurrent presentation, depending on the number of propositions and on the canonical word order, we observed that the differences between both conditions were significant only for canonical sentences (*t*_(100.534)_ = 2.140, *p* = 0.035, *d* = 0.39), albeit they were marginally significant (*p* < 0.10) for sentences with one or two propositions (see Table 1).

Next, we considered how CR and WM accounted for the performance of the participants on CS (Table 5) and NoCS (Table 6), depending on the concurrent or the serial version of the task. We found that CR alone predicted the performance in the concurrent version of both types of sentences. However, when considering the serial version, both CR and WM predicted the performance in CS, and the only variable with predictive power for the comprehension of NoCS was WM. Notably, the percentage of variance explained by these variables tended to be higher in CS (29–34%) than in NoCS (14–16%). Figure 3 and Figure 4 display in detail the unique and shared contributions made by CR, WM, and all the rest of the variables for CS and NoCS, separately.

Moreover, on propositional density, CR alone explained the performance in both concurrent and serial versions of the task for 1PS (Table 7). Conversely, for 2PS, both CR and WM explained the performance in both versions (Table 8). The only difference between the concurrent and serial versions in 2PS was the relative weight of both variables, with a higher weight of CR in the concurrent version and a higher weight of WM in the serial one. Notably, the percentage of variance explained for the models regarding 1PS is slightly lower (14–19%) than the percentage of variance explained for the models regarding 2PS (25–33%). Figure 5 and Figure 6 display in detail the unique and shared contributions made by CR, WM, and all the rest of the variables for 1PS and 2PS, separately.

CR: cognitive reserve, measured by the cognitive reserve questionnaire (CRQ); ß = standardised regression coefficient; TOL: tolerance; VIF: variance inflation factor; *R*^2^: coefficient of determination; Adj *R*^2^: adjusted *R*^2^; Δ*R*^2^: change in *R*^2^ value; Δ*F*: F change; *D*: Durbin–Watson statistic.

Finally, Table 9 presents a summary of the results of the regression analyses for the different measures of sentence comprehension.

## 4. Discussion

The maintenance of language and reading skills is essential for the well-being and quality of life of older adults [44,45]. Thus, in this study, we aimed to explore the following: (a) how the ability to compare linguistic information and information from a visual scene might be affected differently when both sources of information are simultaneously available compared with a situation where verbal information precedes visual information and needs to be kept in memory in order to perform the task; (b) how CR and WM capacity modulate the task performance in these two scenarios; and (c) how this might relate to linguistic complexity at the sentence level. To do so, we compared the performance of two similar groups of cognitively healthy older adults on a sentence–picture verification task involving either the concurrent or the serial presentation of sentence–picture pairs. Moreover, we took measures of CR, WM capacity, and overall cognitive function.

### 4.1. Concurrent vs. Serial Presentation and the Role of WM and CR

Our findings demonstrate that, in general, the serial condition was harder for older adults. Moreover, the performance in the serial version of the task depended on WM capacity to a greater extent than in the concurrent version, as evidenced by the regression analyses.

A seminal study conducted by Waters et al. [26] found no clear effect while comparing a condition where the pictures were presented before the sentences and remained in view to a condition where the sentences and pictures were presented in a serial manner, as in the current experiment. The authors attributed the lack of an effect to the fact that in the serial version, the verbal material was immediately retrieved and no interference occurred. Therefore, they concluded that such a situation posed little additional WM demands. A crucial difference between this early study and the present one is that in the former, the sentences were read by the experimenter, while in the present one, the sentences were available on the screen. Furthermore, when the participants receive verbal auditory input, as in the study by Waters et al., the comparison between verbal and pictorial information might require similar WM demands in both concurrent and serial scenarios. In both situations, the performance depends on the formation of a verbal memory trace, the only difference being the length of time for which this has to remain available. In contrast, in the present study, and in the concurrent version, older adults were allowed to reread while comparing the sentences and the pictures until they responded, thereby effectively minimising WM demands. Compared with this situation, the performance in the serial condition was much more dependent on the encoding and retrieval of an accurate memory trace for linguistic material so as to be able to compare the sentences and the pictures.

On the other hand, the present study confirms the positive role played by CR proxies such as education level, reading habits, occupational status, and other related factors on written sentence comprehension [19,20,25]. The score obtained on the CRQ was positively correlated with sentence reading comprehension in both the concurrent and the serial versions, and it was a main factor in explaining the participants’ performance on the sentence–picture verification task, even after accounting for other interrelated variables in the regression analyses. Recent evidence [20] has shown that high CR might provide efficiency in the processing of syntactic information. Previous research has already shown that print exposure impacts sentence reading among older adults and plays a complementary role in relation to WM capacity [7,46]. The present study supports these findings by using a more comprehensive measure of CR, fashioned across several proxies instead of a single indicator. Moreover, it extends previous evidence, considering together the interplay of CR and WM, depending on the availability of information. While CR alone predicts task performance when the participant is allowed to simultaneously read the sentence and watch the picture, WM becomes an important factor when the sentence and the picture are serially presented.

In regard to the interplay between WM and language comprehension, Caplan and Waters proposed that language interpretive processes (i.e., the processes related to accessing meaning from a linguistic signal) do not depend on the same general-purpose pool of resources involved in the encoding and retrieval of meaning already accessed (or the so-called postinterpretive processes) [47]. According to their findings, age-related reductions in WM resources affect postinterpretive processes but not interpretive processes [2,48,49]. Similarly, Davis et al. [4] demonstrated an age-related bilateral prefrontal modulation of activity related to task demands but not an age-related modulation of the perisylvian language network during sentence processing. Together, these findings suggest that language comprehension and task demands are mediated by related yet different brain networks and that age-related difficulties emerge mainly because of task demands. This idea might have interesting implications for the management and assessment of language and cognitive functions among older adults. On one side, language assessment tasks relying on the encoding and retrieval of verbal material might overestimate language-processing difficulties when compared to the tasks minimising the use of working memory and related executive functions, while on the other side, we should acknowledge that everyday performance in language-mediated tasks might depend on the immediate availability of information in a given context. As suggested by the results of the present study, older adults, particularly those with lower WM resources, might benefit from having verbal information available during the task performance and perform much more poorly when required to retrieve verbal information from memory.

Additionally, the differences in the role played by CR and WM in the concurrent and serial versions also depend on the complexity of the linguistic material. As stated previously, Caplan and Waters differentiated between the processing resources devoted to interpretive processing, which do not seem to be affected by age-related WM depletion, and the postinterpretive resources related to task execution, which would be. Therefore, in the present study, we manipulated two factors related to linguistic complexity, one of which was the adjustment to canonical SVO word order, while the other was the propositional density. Given the difference between interpretive and postinterpretive processes, the adjustment to canonical word order might be considered to influence interpretive processing, while the propositional density should affect postinterpretive processing, as we will discuss in more detail below. Within this framework, the serial presentation, in comparison to the concurrent condition, specifically maximised postinterpretive processing demands.

### 4.2. Concurrent vs. Serial Presentation and Word Order

The word order might influence interpretive processes because of the incrementality of sentence comprehension. Several findings have suggested that we tend to assign more prominent thematic roles such as that of an agent or experiencer to the first animate noun phrases during incremental sentence processing [21]. Therefore, we are forced to reassign thematic roles when this strategy is proven inadequate, as in noncanonical sentences, leading to increased processing demands. Our findings show that WM capacity is not highly relevant to the participants’ performance on the sentence–picture verification task in the concurrent condition for either canonical sentences or noncanonical sentences. Conversely, CR is the main factor that explains the participants’ performance on canonical and noncanonical sentences in the concurrent condition. Recent evidence has linked literacy with the comprehension of complex sentences [50], probably because it favours experience with this kind of material [51]. In this regard, education and reading experience, as contributors to CR, might contribute to facilitating interpretive processing [20]. The situation changes when WM demands increase by the sequential presentation of sentences and pictures. The fact that it happened for both canonical and noncanonical sentences suggests that it is because the serial condition increases the necessity to correctly encode and retrieve information from WM, as discussed above, not because of an interaction between WM capacity and interpretive processing.

Furthermore, we observed a curious and atypical pattern when considering noncanonical sentences. While CR alone was the main predictor in the concurrent version, WM turned out to be the only significant predictor in the serial condition, and CR stopped explaining additional variance in a significant way. This did not happen in the case of any other sentence type, where either CR alone or CR in conjunction with WM contributed to explaining the performance. It has been argued that even if the online interpretive processing of noncanonical sentences is correctly computed, thematic assignment errors increase when the participants have to retrieve information from memory [52,53]. This might happen either because of the difficulties of purging from memory competing misinterpretations on the basis of the linear order of constituents (‘first noun as agent’) or because conflicting cues such as the linear order of words and the sentence structure might interfere when accessing information in a delayed manner. Therefore, even if CR might help in the interpretive processing of noncanonical sentences in the concurrent condition, the misinterpretations based on the linear order of constituents might interfere with memory and supersede any advantages conferred by CR when the testing has been delayed. Additionally, for both concurrent and serial presentations, CR or WM explained little variance in noncanonical sentences. This might be because other relevant variables, such as inhibitory control or cognitive flexibility, were not taken into account. Such variables have been suggested to mediate how older adults cope with the comprehension of complex sentences [54,55,56] and, therefore, might help in selecting from among the conflicting correct and incorrect interpretations arising in noncanonical sentences.

### 4.3. Concurrent vs. Serial Presentation and Propositional Density

In contrast to word order, propositional density is supposed to increase postinterpretive processing demands, and postinterpretive processing is supposed to tap into general WM resources [47]. Semantic information is encoded in a propositional format during sentence comprehension [57]. Hence, the sentences containing higher propositional density provide more information to be encoded in and retrieved from memory [58]. This increases the demands related to operating with the meaning in order to accomplish the task, though not those related to the extraction of the meaning. For example, in a sentence–picture verification task, more information should be compared between the sentence and the picture in two-proposition sentences such as “The dog that bites the horse is big” than in one-proposition sentences such as “It is the dog that bites the horse”. For there to be accurate correspondence between the sentence and picture in the former, a dog should be biting a horse, and the dog must be big, not small. In the second one, any dog biting any horse is enough. The necessity to compare more information during sentence–picture verification might require WM resources even during the supposedly simple concurrent version and much more during the serial version.

Consistent with this, WM capacity (in tandem with CR) turned out to be a relevant variable for the performance in both versions of the task when the sentences had more propositional content (i.e., in two-proposition sentences). Notably, CR was more relevant than WM capacity in the concurrent version with low WM demands, while WM capacity played a more prominent role than CR in the highly demanding serial version.

In contrast, WM capacity did not significantly contribute to the participants’ performance in the sentence–picture verification task for either the concurrent version or the serial version when propositional content was low (i.e., in one-proposition sentences). Therefore, it seems that low WM demands posed by sentences with only one proposition allow the encoding and retrieval of information in the serial version to correctly function even among older adults with low WM capacity.

### 4.4. Limitations

One of the limitations of the present study is that we used single measures for all the relevant variables, with the aim of collecting the data in a single and affordable session. However, it might be interesting to try to replicate the present results by using a larger set of measures for each factor studied, which might allow us to work at the latent variable level. Moreover, we did not consider additional age-related changes in cognition, such as those concerning processing speed (e.g., [59]) or cognitive control (e.g., [56]), which might also play significant roles in the sentence reading comprehension of older adults in conjunction with those studied here.

On the other hand, the current study has separately addressed the demands posed by the adjustment to canonical SVO word order and the propositional density, as they were hypothesised to tap onto interpretive and postinterpretive processing, respectively. Future studies might address possible interactions between canonical word order and propositional density and examine how they are related to WM and CR.

It could be argued that the influence of CR in sentence comprehension comes mainly from a near transfer of reading habits. However, previous results have offered a different perspective. Many previous studies have shown an association of CR with different cognitive domains, including episodic memory, working memory and overall cognitive status [15] (see also [9,33]), and not only language processing. More important, previous research has shown that the contribution of CR to sentence comprehension does not occur directly but rather through the mediatory role of executive functions [19].

## 5. Conclusions

The present study helps us better understand the complementary roles played by CR and WM during sentence reading comprehension among older adults. The factors contributing to CR were related to a better overall performance in our sentence comprehension task. Once CR was taken into account, WM was relevant mainly for the situation where the linguistic and visual materials were presented in a serial fashion and not when verbal and visual information was simultaneously available. An exception to this overall pattern was the result obtained in sentences with high propositional density, where WM was important even when verbal information and the picture were simultaneously presented. This might happen because this kind of sentence imposes higher demands on WM capacity, even if the sentence and the picture are simultaneously available. Moreover, CR did not demonstrate playing positive role on noncanonical sentences when the sentence and the picture were serially presented. Given that noncanonical sentences are prone to misinterpretations and given that the serial presentation exerted hard demands on WM, it is likely that CR might not be able to compensate for the interfering misinterpretations lingering on memory in this complex condition.

## Figures and Tables

**Figure 1 brainsci-13-00428-f001:**
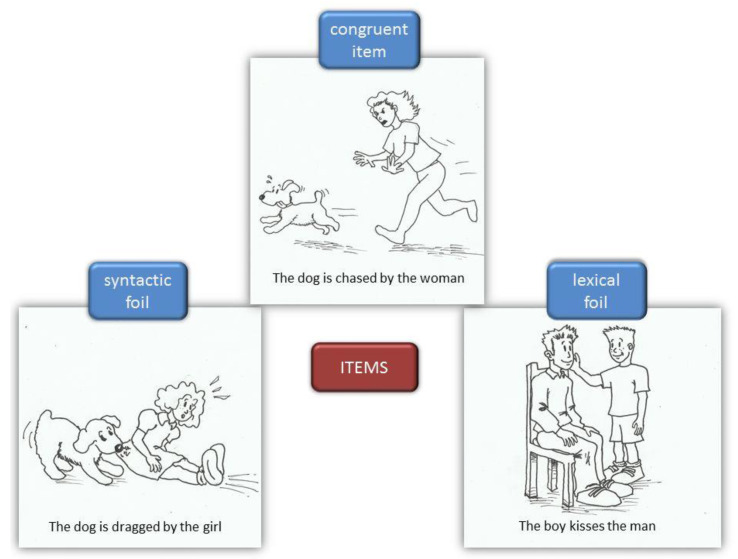
Types of items and examples from ECCO_Senior.

**Figure 2 brainsci-13-00428-f002:**
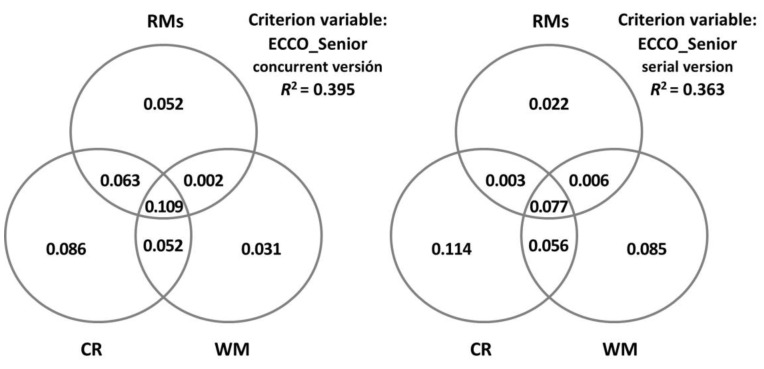
Decomposition of the variance in the ECCO_Senior test explained jointly and uniquely by cognitive reserve (CR), measured by the cognitive reserve questionnaire (CRQ); working memory (WM), measured by the digit ordering task (DO); and the rest of the measures (RMs: age, MMSE, GDS-15, and MFE).

**Figure 3 brainsci-13-00428-f003:**
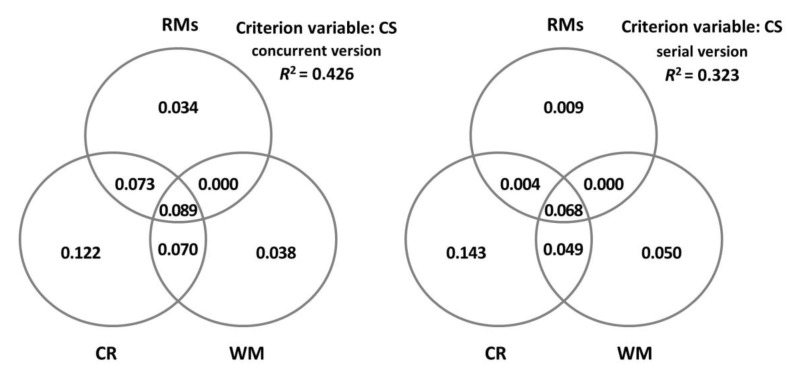
Decomposition of the variance in canonical sentences (CSs) explained jointly and uniquely by cognitive reserve (CR), measured by the cognitive reserve questionnaire (CRQ); working memory (WM), measured by the digit ordering task (DO); and the rest of the measures (RMs: age, MMSE, GDS-15, and MFE).

**Figure 4 brainsci-13-00428-f004:**
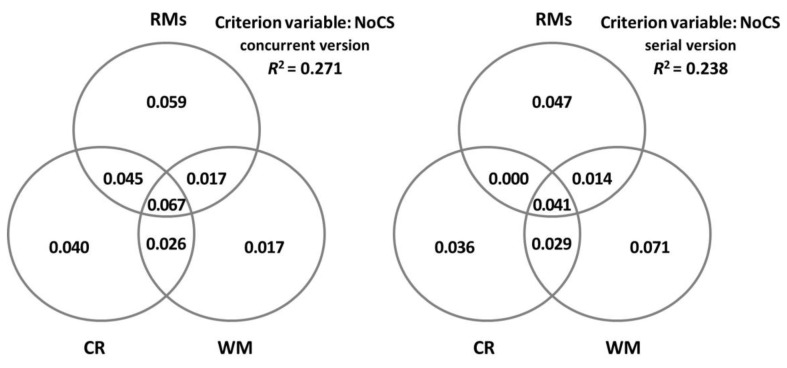
Decomposition of the variance in noncanonical sentences (NoCS) explained jointly and uniquely by cognitive reserve (CR), measured by the cognitive reserve questionnaire (CRQ); working memory (WM), measured by the digit ordering task (DO); and the rest of the measures (RMs: age, MMSE, GDS-15, and MFE).

**Figure 5 brainsci-13-00428-f005:**
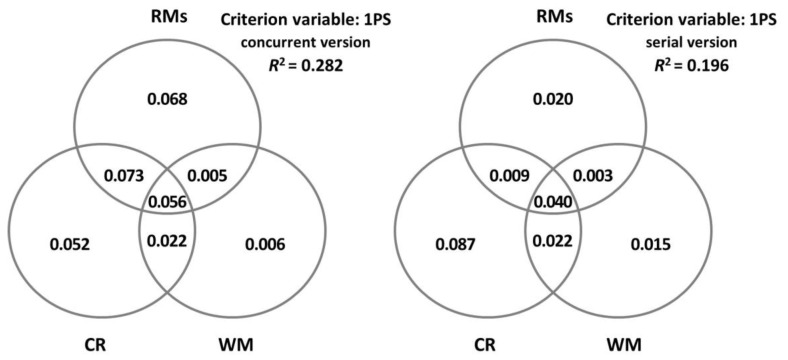
Decomposition of the variance in one-proposition sentences (1PS) explained jointly and uniquely by cognitive reserve (CR), measured by the cognitive reserve questionnaire (CRQ); working memory (WM), measured by the digit ordering task (DO); and the rest of the measures (RMs: age, MMSE, GDS-15, and MFE).

**Figure 6 brainsci-13-00428-f006:**
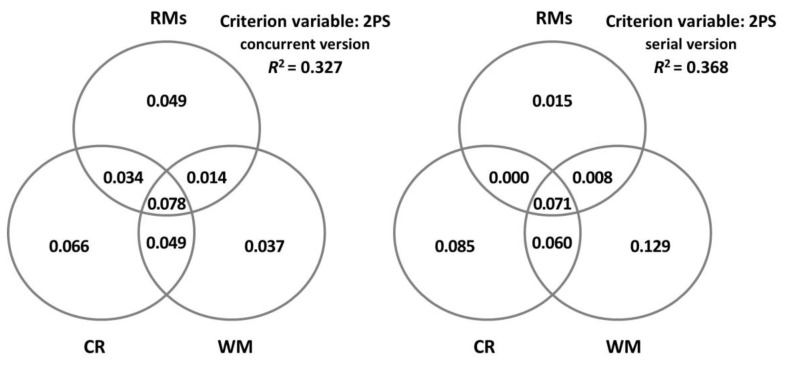
Decomposition of the variance in two-proposition sentences (2PS) explained jointly and uniquely by cognitive reserve (CR), measured by the cognitive reserve questionnaire (CRQ); working memory (WM), measured by the digit ordering task (DO); and the rest of the measures (RMs: age, MMSE, GDS-15, and MFE).

**Table 1 brainsci-13-00428-t001:** Descriptive statistics and mean differences between groups for all variables.

	Concurrent Version (*n* = 60)						Serial Version (*n* = 60)						Contrast		
	*S*	*K*	Min	Max	*M*	SD	*S*	*K*	Min	Max	*M*	SD	*t* _(118)_	*p*	*d*
Age	0.08	−0.80	60	80	69.23	5.28	0.19	−0.57	60	80	69.02	4.98	0.231	0.818	-
MMSE (max = 30)	−1.30	0.56	27	30	29.33	0.97	−1.38	0.69	27	30	29.37	0.97	−0.188	0.851	-
GDS-15 (max = 15)	0. 51	−0.55	0	5	1.70	1.47	0.55	−0.85	0	5	1.63	1.61	0.238	0.813	-
MFE (max = 56)	0.18	−0.37	0	13	6.22	2.74	0.38	−0.65	0	13	5.43	3.22	1.435	0.154	-
CRQ (max = 25)	−0.38	−0.38	3	22	13.98	4.53	0.03	−0.50	7	24	13.95	4.04	0.043	0.966	-
DO (max = 15)	−0.90	0.28	7	15	12.38	1.91	−0.43	−0.26	7	15	11.92	1.96	1.320	0.190	-
ECCO_Senior (max = 36)	−0.63	−0.32	24	36	31.38	2.92	−0.59	−0.53	23	35	30.13	3.35	2.179	**0.031**	0.40
CS (max = 18)	−1.15	0.65	13	18	16.75	1.27	−1.02	0.12	11	18	16.10	1.98	2.140 ^a^	**0.035**	0.39
NoCS (max = 18)	−0.56	−0.01	9	18	14.63	2.05	−0.19	−0.75	10	18	14.03	2.01	1.620	0.108	0.30
1PS (max = 18)	−1.10	1.96	11	18	16.17	1.43	−0.91	0.37	11	18	15.62	1.72	1.907	*0.059*	0.35
2PS (max = 18)	−0.52	−0.58	11	18	15.22	2.06	−0.27	−0.80	10	18	14.52	2.14	1.828	*0.070*	0.33

MMSE: mini-mental state examination; GDS-15: the short form of the geriatric depression scale; MFE: memory failures of everyday questionnaire; CRQ: cognitive reserve questionnaire; DO: digit ordering; ECCO_Senior: the Spanish test evaluación cognitiva de la comprensión de oraciones para mayores (sentence–picture verification task). CS: canonical sentences (part of the ECCO_Senior test); NoCS: noncanonical sentences (part of the ECCO_Senior test); 1PS: one-proposition sentences (part of the ECCO_Senior test); 2PS: two-proposition sentences (part of the ECCO_Senior test); *S*: skewness; *K*: kurtosis; Min: minimum; Max: maximum; *M*: mean; SD: standard deviation; *t*: *t*-test values; *p*: *p*-values for two-tailed *t*-test; *d*: Cohen’s *d*-effect size. Boldface indicates significant difference (*p* < 0.05); italics indicate marginally significant difference (0.05 > *p* < 0.10); ^a^: Levene’s test indicates heteroscedastic variances (*F* = 11.568, *p* = 0.001). Significant *p*-values: *p* < 0.05.

**Table 2 brainsci-13-00428-t002:** Sentence structure types included in the ECCO_Senior test.

Sentence Structure	Canonical Word Order	Number of Propositions	Example
Active	Yes	1	El caballo mordió al perro (Spanish) *The horse bit the dog (English)*
Passive	No	1	El hombre es adelantado por el caballo *The man is passed by the horse*
Cleft subject	Yes	1	Es el perro el que mordió al gato *It is the dog that bites the cat*
Cleft object	No	1	Es a la mujer a la que despierta el hombre *It is the woman that the man wakes up*
V-PrepP-NP passive	Yes	1	Es despertado por el hombre el niño *The boy is awakened by the man*
V-NP-PrepP passive	No	1	Es atacado el gato por el niño *The cat is attacked by the boy*
Subject relative clause, present continuous	Yes	2	El perro que está arrastrando al gato es pequeño. *The dog that is dragging the cat is small*
Object relative clause, present continuous	No	2	El gato que el caballo está persiguiendo es blanco. *The cat that the horse is chasing is white*
Subject relative clause	Yes	2	El perro que mordió al caballo es grande. *The dog that bit the horse is big*
Object relative clause	No	2	El perro al que el niño arrastró es pequeño. *The dog that the boy dragged is small*
Object–subject relative clause	Yes	2	El niño besó a la mujer que arrastra al perro. *The boy kissed the woman who drags the dog*
Subject–object relative clause	No	2	El perro al que el gato mordió empuja al niño. *The dog that the cat bit pushes the boy*

**Table 3 brainsci-13-00428-t003:** Correlation matrix by group: coefficients for concurrent and serial conditions on the upper and lower triangles, respectively.

Variable		1	2	3	4	5	6	7	8	9	10	11
1	Age		**−0.31**	−0.16	0.04	−0.21	−0.21	−0.24	−0.14	−0.26	−0.07	**−0.30**
2	MMSE	**−0.35**		−0.13	−0.08	**0.35**	**0.40**	**0.27**	0.23	0.23	0.24	0.21
3	GDS-15	−0.12	−0.21		0.11	**−0.36**	−0.21	**−0.35**	**−0.34**	**−0.28**	**−0.38**	−0.22
4	MFE	−0.09	0.11	0.04		0.01	**−0.13**	−0.05	0.09	−0.13	0.06	−0.12
5	CRQ	−0.20	**0.44**	−0.25	−0.02		**0.46**	**0.56**	**0.60**	**0.42**	**0.45**	**0.48**
6	DO	**−0.34**	**0.50**	−0.09	0.02	**0.40**		**0.44**	**0.44**	**0.36**	**0.30**	**0.42**
7	ECCO_Senior	−0.22	**0.28**	−0.11	0.12	**0.50**	**0.48**		**0.80**	**0.93**	**0.76**	**0.89**
8	CS	−0.18	0.24	−0.12	−0.02	**0.52**	**0.41**	**0.83**		**0.52**	**0.69**	**0.66**
9	NoCS	−0.18	0.24	−0.05	0.22	**0.33**	**0.40**	**0.84**	**0.41**		**0.66**	**0.86**
10	1PS	−0.14	0.22	−0.11	0.13	**0.40**	**0.28**	**0.83**	**0.70**	**0.70**		**0.39**
11	2PS	−0.22	**0.27**	−0.08	0.08	**0.47**	**0.53**	**0.90**	**0.75**	**0.76**	**0.50**	

MMSE: mini-mental state examination; GDS-15: the short form of the geriatric depression scale; MFE: memory failures of everyday questionnaire; CRQ: cognitive reserve questionnaire; DO: digit ordering; ECCO_Senior: the Spanish test evaluación cognitiva de la comprensión de oraciones para mayores (sentence–picture verification task). CS: canonical sentences (part of the ECCO_Senior test); NoCS: noncanonical sentences (part of the ECCO_Senior test); 1PS: one-proposition sentences (part of the ECCO_Senior test); 2PS: two-proposition sentences (part of the ECCO_Senior test). Significant correlations are displayed in bold: *r*s > |±0.27| significant at the level of *p* < 0.05; *r*s > |±0.34| significant at the level of *p* < 0.01; *r*s > |±0.44| significant at the level of *p* < 0.001.

**Table 4 brainsci-13-00428-t004:** Results of the stepwise regression analyses for ECCO_Senior.

	ß	*t*	*p*	TOL	VIF	*R* ^2^	Adj *R*^2^	Δ*R*^2^	Δ*F*	df	*p*	*D*
Concurrent condition												
Model 1						0.309	0.297	0.309	25.880	1, 58	<0.001	1.999
CR	0.555	5.087	<0.001	1.000	1.000							
Serial condition												
Model 1						0.251	0.238	0.251	19.402	1, 58	<0.001	
CR	0.501	4.405	<0.001	1.000	1.000							
Model 2						0.344	0.321	0.093	8.082	1, 57	0.006	1.939
CR	0.368	3.140	0.003	0.840	1.190							
WM	0.333	2.843	0.006	0.840	1.190							

CR: cognitive reserve, measured by the cognitive reserve questionnaire (CRQ); WM: working memory, measured by the digit ordering task (DO); ß = standardised regression coefficient; TOL: tolerance; VIF: variance inflation factor; *R*^2^: coefficient of determination; Adj *R*^2^: adjusted *R*^2^; Δ*R*^2^: change in *R*^2^ value; Δ*F*: *F* change; *D*: Durbin–Watson statistic.

**Table 5 brainsci-13-00428-t005:** Results of the stepwise regression analyses for canonical sentences.

	ß	*t*	*p*	TOL	VIF	*R* ^2^	Adj *R*^2^	Δ*R*^2^	Δ*F*	df	*p*	*D*
Concurrent condition												
Model 1						0.356	0.345	0.356	32.034	1, 58	<0.001	2.178
CR	0.596	5.660	<0.001	1.000	1.000							
Serial condition												
Model 1						0.266	0.253	0.266	20.985	1, 58	<0.001	
CR	0.515	4.581	<0.001	1.000	1.000							
Model 2						0.314	0.290	0.049	4.048	1, 57	0.049	1.907
CR	0.419	3.503	0.001	0.840	1.190							
WM	0.241	2.012	0.049	0.840	1.190							

CR: cognitive reserve, measured by the cognitive reserve questionnaire (CRQ); WM: working memory, measured by the digit ordering task (DO); ß = standardised regression coefficient; TOL: tolerance; VIF: variance inflation factor; *R*^2^: coefficient of determination; Adj *R*^2^: adjusted *R*^2^; Δ*R*^2^: change in *R*^2^ value; Δ*F*: *F* change; *D*: Durbin–Watson statistic.

**Table 6 brainsci-13-00428-t006:** Results of the stepwise regression analyses for noncanonical sentences.

	ß	*t*	*p*	TOL	VIF	*R* ^2^	Adj *R*^2^	Δ*R*^2^	Δ*F*	df	*p*	*D*
Concurrent condition												
Model 1						0.178	0.164	0.178	12.595	1, 58	0.001	1.951
CR	0.422	3.549	0.001	1.000	1.000							
Serial condition												
Model 1						0.158	0.143	0.158	10.848	1, 58	0.002	1.652
WM	0.397	3.294	0.002	1.000	1.000							

CR: cognitive reserve, measured by the cognitive reserve questionnaire (CRQ); WM: working memory, measured by the digit ordering task (DO); ß = standardised regression coefficient; TOL: tolerance; VIF: variance inflation factor; *R*^2^: coefficient of determination; Adj *R*^2^: adjusted *R*^2^; Δ*R*^2^: change in *R*^2^ value; Δ*F*: *F* change; *D*: Durbin–Watson statistic.

**Table 7 brainsci-13-00428-t007:** Results of the stepwise regression analyses for one-proposition sentences.

	ß	*t*	*p*	TOL	VIF	*R* ^2^	Adj *R*^2^	Δ*R*^2^	Δ*F*	df	*p*	*D*
Concurrent condition												
Model 1						0.203	0.189	0.203	14.780	1, 58	<0.001	1.825
CR	0.451	3.844	<0.001	1.000	1.000							
Serial condition												
Model 1						0.158	0.144	0.158	10.894	1, 58	0.002	2.144
CR	0.398	3.301	0.002	1.000	1.000							

**Table 8 brainsci-13-00428-t008:** Results of the stepwise regression analyses for two-proposition sentences.

	ß	*t*	*p*	TOL	VIF	*R* ^2^	Adj *R*^2^	Δ*R*^2^	Δ*F*	df	*p*	*D*
Concurrent condition												
Model 1						0.227	0.213	0.227	16.991	1, 58	<0.001	
CR	0.476	4.122	<0.001	1.000	1.000							
Model 2						0.279	0.253	0.052	4.115	1, 57	0.047	2.060
CR	0.358	2.825	0.007	0.789	1.268							
WM	0.257	2.029	0.047	0.789	1.268							
Serial condition												
Model 1						0.276	0.263	0.276	22.057	1, 58	<0.001	
WM	0.525	4.696	<0.001	1.000	1.000							
Model 2						0.353	0.330	0.077	6.815	1, 57	0.012	1.719
WM	0.404	3.472	0.001	0.840	1.190							
CR	0.303	2.611	0.012	0.840	1.190							

CR: cognitive reserve, measured by the cognitive reserve questionnaire (CRQ); WM: working memory, measured by the digit ordering task (DO); ß = standardised regression coefficient; TOL: tolerance; VIF: variance inflation factor; *R*^2^: coefficient of determination; Adj *R*^2^: adjusted *R*^2^; Δ*R*^2^: change in *R*^2^ value; Δ*F*: *F* change; *D*: Durbin–Watson statistic.

**Table 9 brainsci-13-00428-t009:** Summary of results from the regression analyses.

Outcome Measure	Group	Significant Predictors
ECCO_Senior	Concurrent	CR
	Serial	CR, WM
CS	Concurrent	CR
	Serial	CR, WM
NoCS	Concurrent	CR
	Serial	WM
1PS	Concurrent	CR
	Serial	CR
2PS	Concurrent	CR, WM
	Serial	WM, CR

ECCO_Senior: The Spanish test evaluación cognitiva de la comprensión de oraciones para mayores (sentence–picture verification task).CS: canonical sentences (part of the ECCO_Senior test); NoCS: noncanonical sentences (part of the ECCO_Senior test); 1PS: one-proposition sentences (part of the ECCO_Senior test); 2PS: two-proposition sentences (part of the ECCO_Senior test); CR: cognitive reserve, measured by the cognitive reserve questionnaire (CRQ); WM: working memory, measured by the digit ordering task (DO).

## Data Availability

All the data obtained in this study are available upon request to gmejuto@lagunacuida.org or rlopezsa@psi.ucm.es.

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
