# Peer review of "Task Demands and Sentence Reading Comprehension among Healthy Older Adults: The Complementary Roles of Cognitive Reserve and Working Memory"

_brainsci, 2023, doi:10.3390/brainsci13030428_

Round 1

Reviewer 1 Report

This study investigates the interplay of cognitive reserve and working memory using a protocol in which healthy older adults (aged 60-80) completed a sentence-picture matching task.  The parameters that were manipulated include manner of presentation of the sentences and pictures, either concurrent or serial; what they call “canonical” vs. “noncanonical” word order, of which there are 3 different sentence structures for each type of word order; and 1 vs. 2 propositions for each type of sentence structure.  This design yields 12 sentence types.  For each sentence type, there were 3 pictures.  In one, the picture matched the sentence, in one there was a discrepancy between the sentence and the picture due to a syntactic mismatch, and in one there was a discrepancy due to a lexical mismatch.  Thus there were 36 items altogether.

The experimental design examined two predictor variables, cognitive reserve (CR) and working memory (WM).  Throughout the paper, cognitive reserve is variously referred to as CR, CR proxies, and CRQ, for “cognitive reserve questionnaire (score).”  This is somewhat confusing and should be simplified for consistency.  Once it’s established that the CRQ was used to assess CR and that the questionnaire probed proxies for cognitive reserve such as education and occupation, then only one label is needed, and it should be used consistently.  Working memory is referred to as WM and DO, for “digit ordering (score),” the digit span test that was used to assess working memory.  Again, this should be simplified to just one label.

The outcome variable was the score on the 36-item sentence-matching test described above, the ECCO_Senior test.  In an odd oversight, Section 2.3, describing the experimental procedure, does not say that this test was administered; it only says that the participants were assigned to groups for the concurrent vs. serial presentation of the test.  It does say that the tasks for assessing CR and WM were administered, but it should also say explicitly that the participants completed the ECCO_Senior test.

Table 3 absolutely must have labels at the tops of the columns.  It is not at all obvious that the 1, 2, 3, etc. across the top of the table refer to the numbering of the rows, and even when the reader figures that out, it is difficult read the table without explanatory headings.  Also, the labels “concurrent” and “serial” should be added to the respective sections of the table, placed in the body of the table.

Lines 314 ff: It is confusing that the authors specify age as a variable that did not explain a significant portion of the variance in the regression analysis.  This can be deduced from Table 4 by its absence, but it would be much simpler to state this plainly and explain that the same is true of the other global assessment factors, namely MMSE, GDS-15, and MFE. 

The Venn diagrams in Figures 2-6 are interesting, but one wonders how they were arrived at.  There is no explanation of how the variance in the ECCO_Senior test was decomposed into components of separate and overlapping contributions to the total variance.  This is a major omission that needs to be corrected.

The results are complicated and difficult to keep track of.  It would help the reader to include a simple table along the lines of the following:

Sentence Type Condition       Significant Predictors

all                    concurrent      CR

                         serial              CR, WM

canonical        concurrent      CR

                         serial              CR, WM

non-canonical concurrent      CR

                         serial              WM

1 proposition  concurrent      CR

                         serial              CR

2 propositions concurrent      CR, WM

                         serial              WM, CR

The results of this experiment overall are not surprising and don’t seem to break new ground.  It would be interesting to know more of the details of intersecting categories, such as canonical word order with 1 proposition in the concurrent and serial conditions vs. canonical word order with 2 propositions in the 2 conditions.  In general, the participants performed fairly close to ceiling, and it would be worthwhile to see which combinations of categories were real impediments to performance.  Also, it would be valuable to compare these findings for older adults with findings for young adults, since the participants performed so well.  The authors imply that older adults experience declining WM, but they do not provide evidence of decline.  The maximum score attainable for each variable should be given in Table 1, as in “MMSE (max=30).”  Norms for digit span and the cognitive reserve questionnaire for young adults could be provided in the notes to that table.

In Table 2, the categories given as “Subject passivized relative clause” and “Object passivized relative clause” are incorrectly labeled.  There are no passives in these sentences; “passivized” should be removed from the category labels.  In the labels for the next two categories, “Subject embedded relative clause” and “Object embedded relative clause,” the word “embedded” is redundant and should be removed.  I don’t know how to interpret the labels “V-PrepP-NP passive” and “V-NP-PrepP passive,” but at least these sentences are passives.

The paper is generally adequately well-written, but there are some features that should be corrected through careful review by a native speaker of English.  “Performance,” in the sense of score on a test, should not be preceded by the definite article.  Thus, for example, line 290 should read, “Age was not significantly related to performance in either of the two…”  There are numerous cases of poor word choice, such as “widely intertwined” (line 35); “a higher extent” (line 139; this should be “a greater extent”); “constituting 36 sentence-picture pairs” (line 195; should be “consisting of”), and others.  In line 238, the authors write “After the exclusion criteria were ruled out…”  They mean that some prospective participants were ruled out by the exclusion criteria, but the sentence as it stands does not convey that.

Author Response

Dear Reviewer,

Please, find our reply in attach file.

Best regards.

Reviewer 2 Report

This is an interesting and well-done study that examines the relationships between activities assumed to promote cognitive reserve (CRQ task), working memory capacity (WMC) and performance on a sentence-picture verification task that varied in terms of canonical word order and/or number of propositions.  In addition, the WM demands of the sentence-picture task were manipulated between subjects by using either concurrent presentation (low load) or by presenting the sentence before the picture (high load).

Higher CRQ scores were associated with better performance in both sentence-picture conditions, but higher WMC benefited sentence-picture performance only when the presentation was sequential and especially for the more complex sentences.

Although the overall pattern of results makes sense, I find the most intriguing result to be that the CRQ scores are good predictors (e.g., beta > .501) of both the concurrent and sequential conditions of the picture-sentence task and that makes me want to know more about the CRQ scale. The authors helpfully state that it “covers 8 areas that have been associated with the formation of CR: education, parents’ education, continuing education, occupation, musical training, knowledge of co-official and/or foreign language(s), reading activity, and engagement in intellectual games.” Unfortunately, the cited publication (Rami, et al., 2011) is in Spanish and many interested readers of this study will not be able to readily access any information about the validity and reliability of the CRQ. I am also interested in knowing how many separable factors emerge from the “8 areas” and whether it would be worthwhile to also report subscale analyses.  I also feel a bit uneasy about the construct captured by the CRQ and what a “cognitive reserve proxy” can tell us.  To take one concrete example, one of the activities is “reading activity”, an activity that through very near transfer should enhance performance on the picture-sentence task.  Any contribution to the betas mentioned above from the “reading activity” may have little or nothing to do with any mediating effects on brain structure that support general cognitive ability.  Is there an effect above and beyond the real possibility that individuals who read a lot are better at the picture-sentence verification task? Similar questions could be asked about some of the other activities such as education.

Author Response

(The authors gave the same response as above.)

Reviewer 3 Report

A better definition of cognitive reserve and the relationship with cognitive performance (in a task of WM) would be appreciated.

The cognitive reserve must not always be described in terms of resilience and help when there is difficulty/impairment, but also as a boost, a cumulative capacity acquired in case of high cognitive tasks in healthy participants

1. In Table 1 all listed tests and questionnaires have to be appropriately quoted .... not all of them are reported. All acronyms have to be  explained

2. Montemurro et al. 2021 must be quoted:  https://doi.org/10.1080/23273798.2021.1896012 

3. What was the hypothesis for the variable sex at the beginning?

The manuscript it is well written but less acronyms would be beneficial for the readability of the text

Author Response

Dear Reviewer,

Please find our reply in attach file.

Best regards.

Round 2

Reviewer 1 Report

This is a review of the first revision of a study that investigates the interplay of cognitive reserve and working memory using a protocol in which healthy older adults completed a sentence-picture matching task.  On the whole, the authors have responded satisfactorily to my earlier comments.  I can offer a few more observations.

The authors agree that it is appropriate to use one term consistently for cognitive reserve and for working memory, and they use CR and WM, respectively, in the text.  However, they continue to use CRQ and DO, respectively, in the tables (3-9) and figures (2-5), even when the text directly refers to the results shown in the tables and figures.  This mismatch is confusing and unnecessary.  If the authors feel they need to state explicitly that “CR” and “WM” in the tables and figures stand for the participants’ scores on the corresponding tasks, then they can do so, but the references in the text should match the references in the tables and figures.

I do not agree that Table 3, showing the cross-correlation matrix of demographic variables and test scores, is clear.  If the authors do not want to use headings across the top of the table that match the variable names, at least they can move the label “Variable” to the left so that it is above the list of numbers, 1, 2, 3, etc., up to 11, to help the reader understand that the numbers across the top correspond to the numbers, and hence the variables, in the list.

I agree that the issue of intersecting categories, such as canonical word order with one proposition in the concurrent and serial conditions vs. canonical word order with 2 propositions in the 2 conditions invites further study.  This could be mentioned in the “Limitations” section.

As I said in my earlier remarks, the authors imply that older adults experience declining working memory, but they do not provide evidence of decline.  Indeed, there seems to be little evidence of variability among the participants in this study.  Can the authors comment on the amount of variability?  It would be informative to have some kind of norms with which to compare the performance of the participants in this study.

The test sentences of the types V-PrepP-NP passive and V-NP-PrepP passive would be very strange in English: “Is attacked the cat by the boy.”  If this word order is normal in Spanish, it would be appropriate to state that.

Minor comments on lexicon and grammar: 

p. 5, line 195: “on people with AD…” should be “in people with AD…”

p. 7, line 261: “They…” should be “The participants…”

p. 15, line 494:  “evidence have shown…” should be “evidence has shown…”

Author Response

Dear Reviewer,

Thank you for your suggestions/comments.

Please find in attach, in blue font, response to your comments. We believe these changes have strengthened the manuscript.

We hope that the revised manuscript now meets the high standards of Brain Sciences, and we look forward to your feedback on our submission.

Thank you again for your time and consideration.
